# Antidepressant Use and Its Association with 28-Day Mortality in Inpatients with SARS-CoV-2: Support for the FIASMA Model against COVID-19

**DOI:** 10.3390/jcm11195882

**Published:** 2022-10-05

**Authors:** Nicolas Hoertel, Marina Sánchez-Rico, Johannes Kornhuber, Erich Gulbins, Angela M. Reiersen, Eric J. Lenze, Bradley A. Fritz, Farid Jalali, Edward J. Mills, Céline Cougoule, Alexander Carpinteiro, Christiane Mühle, Katrin Anne Becker, David R. Boulware, Carlos Blanco, Jesús M. Alvarado, Nathalie Strub-Wourgaft, Cédric Lemogne, Frédéric Limosin

**Affiliations:** 1Institut de Psychiatrie et Neuroscience de Paris, Université Paris Cité, INSERM U1266, F-75014 Paris, France; 2AP-HP, DMU Psychiatrie et Addictologie, Hôpital Corentin-Celton, Issy-les-Moulineaux, F-92130 Paris, France; 3Department of Psychobiology and Behavioural Sciences Methods, Faculty of Psychology, Universidad Complutense de Madrid, Pozuelo de Alarcón, 28223 Pozuelo de Alarcón (Madrid), Spain; 4Department of Psychiatry and Psychotherapy, University Hospital, Friedrich-Alexander-University of Erlangen-Nuremberg (FAU), 91054 Erlangen, Germany; 5Institute for Molecular Biology, University Hospital Essen, University of Duisburg-Essen, 47057 Essen, Germany; 6Department of Psychiatry, Washington University School of Medicine, St. Louis, MO 63110, USA; 7Department of Anesthesiology, Washington University School of Medicine, St. Louis, MO 63110, USA; 8Department of Gastroenterology, Saddleback Medical Group, Laguna Hills, CA 92653, USA; 9Health Research Methods, Evidence, and Impact, McMaster University, Hamilton, ON L8S 4K1, Canada; 10Institut de Pharmacologie et de Biologie Structurale (IPBS), Université de Toulouse, F-31400 Toulouse, France; 11Department of Hematology and Stem Cell Transplantation, University Hospital Essen, University of Duisburg-Essen, 47057 Essen, Germany; 12Department of Medicine, Division of Infectious Diseases and International Medicine, University of Minnesota, Minneapolis, MN 55455, USA; 13National Institute on Drug Abuse (NIDA), National Institutes of Health, Bethesda, MD 20852, USA; 14COVID-19 Response & Pandemic Preparedness, Drugs for Neglected Diseases Initiative (DNDi), 1202 Geneva, Switzerland; 15Service de Psychiatrie de l’adulte, AP-HP, Hôpital Hôtel-Dieu, DMU Psychiatrie et Addictologie, F-75004 Paris, France

**Keywords:** antidepressant, fluoxetine, fluvoxamine, COVID-19, SARS-CoV-2, mortality, sphingomyelinase, ceramide, FIASMA, sigma-1 receptor

## Abstract

To reduce Coronavirus Disease 2019 (COVID-19)-related mortality and morbidity, widely available oral COVID-19 treatments are urgently needed. Certain antidepressants, such as fluvoxamine or fluoxetine, may be beneficial against COVID-19. We included 388,945 adult inpatients who tested positive for SARS-CoV-2 at 36 AP–HP (Assistance Publique–Hôpitaux de Paris) hospitals from 2 May 2020 to 2 November 2021. We compared the prevalence of antidepressant use at admission in a 1:1 ratio matched analytic sample with and without COVID-19 (N = 82,586), and assessed its association with 28-day all-cause mortality in a 1:1 ratio matched analytic sample of COVID-19 inpatients with and without antidepressant use at admission (N = 1482). Antidepressant use was significantly less prevalent in inpatients with COVID-19 than in a matched control group of inpatients without COVID-19 (1.9% versus 4.8%; Odds Ratio (OR) = 0.38; 95%CI = 0.35–0.41, *p* < 0.001). Antidepressant use was significantly associated with reduced 28-day mortality among COVID-19 inpatients (12.8% versus 21.2%; OR = 0.55; 95%CI = 0.41–0.72, *p* < 0.001), particularly at daily doses of at least 40 mg fluoxetine equivalents. Antidepressants with high FIASMA (Functional Inhibitors of Acid Sphingomyelinase) activity seem to drive both associations. These treatments may reduce SARS-CoV-2 infections and COVID-19-related mortality in inpatients, and may be appropriate for prophylaxis and/or COVID-19 therapy for outpatients or inpatients.

## 1. Introduction

The global spread of the different variants of Severe Acute Respiratory Syndrome Coronavirus 2 (SARS-CoV-2) has led to an infectious disease crisis worldwide [1,2,3,4]. Because a large proportion of the world’s population is currently unvaccinated, effective treatments of Coronavirus Disease 2019 (COVID-19)—especially those that can be administered orally, have good tolerability and low rate of medical contraindications [5], are inexpensive and immediately available—are urgently needed to reduce COVID-19-related mortality and morbidity [6]. This is particularly important in low- and middle-income countries, where access to vaccines and approved treatments against COVID-19 is limited [7].

Several lines of research suggest that certain well-tolerated [8,9] antidepressants, especially the Selective Serotonin Reuptake Inhibitor (SSRI) medications fluvoxamine or fluoxetine, could be beneficial against COVID-19, and thus a potential means of reaching this goal [7,10,11,12]. Firstly, several preclinical studies have demonstrated in vitro efficacy of several SSRI and non-SSRI antidepressants—particularly fluoxetine—against different variants of SARS-CoV-2 in human and non-human host cells [12,13,14,15,16,17,18]. Secondly, a retrospective cohort study conducted in an adult psychiatric facility suggested a significant negative association of antidepressant use—particularly fluoxetine—with laboratory-detectable SARS-CoV-2 infection [19]. Thirdly, in the ambulatory setting, three studies [20,21,22], including two randomized, placebo-controlled trials (RCT) [20,22] and one non-randomized open-label clinical study [21] found a significant association between the short-term use (10–15 days) of fluvoxamine within 7 days of symptom onset and reduced risk of clinical deterioration. Contrariwise, an RCT of fluvoxamine [23] prescribed at 100 mg/d among over-weighted and obese outpatients with COVID-19 showed no significant benefit on the risk of emergency department visits, hospitalizations, or death, contrasting with the findings of TOGETHER and STOP-COVID, in which fluvoxamine was prescribed at a dose of 200 and 300 mg/d, respectively. Fourthly, an observational study found that exposure to antidepressants, especially to those that functionally inhibit acid sphingomyelinase, was associated with reduced incidence of emergency department visitation or hospital admission among SARS-CoV-2 positive outpatients, in a dose-dependent manner and from daily doses of at least 20 mg fluoxetine equivalents [24]. Fifthly, five retrospective observational cohort studies [25,26,27,28,29] of patients with COVID-19 in the acute-care setting reported reduced death or mechanical ventilation in those taking SSRIs, particularly fluoxetine. Of these five studies, two [25,29] reported a similar association in those taking non-SSRI antidepressants, particularly mirtazapine and venlafaxine. Finally, a prospective cohort study of patients admitted to the intensive care unit (ICU) for COVID-19 reported a significant association between the use of fluvoxamine for 15 days and reduced mortality [30]. Altogether, these findings suggest that the use of certain antidepressants, when prescribed at a dose of at least 20 mg fluoxetine equivalents, may reduce the clinical deterioration of patients infected with SARS-CoV-2 in both ambulatory and acute-care settings.

However, most prior studies focused on a limited number of antidepressant molecules (e.g., only SSRIs). In addition, several of these studies used composite outcomes, such as intubation or death [25,28], which may prove challenging for the interpretation of results. Finally, it remains unknown whether the potential benefit of certain antidepressants in COVID-19 in the acute-care setting is dose-dependent and only observed beyond a certain dose threshold. This knowledge is important to help determine the best drug candidates and their optimal dosing for future clinical trials, as well as to progress in the identification of the mechanisms underlying this potential effect.

To address these knowledge gaps, we used the Assistance Publique–Hôpitaux de Paris (AP–HP) Health Data Warehouse (‘Entrepôt de Données de Santé’ (EDS)) [25,28,29,31,32,33,34,35,36,37,38], which includes data on all adult inpatients aged 18 years or over who had been admitted to any of the 36 AP–HP Greater Paris University hospitals and tested for SARS-CoV-2 infection by a Reverse Transcription Polymerase Chain Reaction (RT-PCR) test at their admission, and performed a large (N = 388,945) multicenter retrospective cohort study.

In this study, our primary aim was two-fold: (i) to test the hypothesis that the prevalence of antidepressant use in patients hospitalized with COVID-19 would be lower than in patients with similar characteristics hospitalized without COVID-19, and (ii) to examine, among patients hospitalized with COVID-19, whether antidepressant use is associated with reduced 28-day mortality. Our secondary aim was to examine whether this potential association could only concern specific antidepressant classes or molecules, is dose-dependent, and/or only observed beyond a certain dose threshold.

## 2. Materials and Methods

### 2.1. Setting and Cohort Assembly

This study included 388,945 hospitalized adult patients who were admitted to AP–HP Greater Paris University hospitals between 2 May 2020 and 2 November 2021, and had been tested for COVID-19 by an RT-PCR test at admission. The sample in this study did not overlap with that of a previous study using the AP–HP Warehouse data [25], which was based on a different inclusion period (i.e., from February 24th to May 1st). AP–HP clinical Data Warehouse initiatives ensure patient information and informed consent regarding the different approved studies through a transparency portal in accordance with European Regulation on data protection and authorization n°1980120 from National Commission for Information Technology and Civil Liberties (CNIL). This observational study using routinely collected data received full approval from the Institutional Review Board (IRB) of the AP–HP clinical data warehouse (decision CSE-20-20_COVID19, IRB00011591). In accordance with French laws for this type of observational non-interventional research study (“reference methodology MR-005”), patients were informed that their data could be used for research, but patient consent was not applicable, as this study did not contain factors necessitating it. Data were anonymized by the AP–HP clinical data warehouse team prior to the analyses.

### 2.2. Variables Assessed

We extracted data from the electronic health record for each patient at the time of the hospitalization regarding patient demographic characteristics, hospitalization dates, laboratory test and RT-PCR test results, medication lists and medication administration data, International Classification of Diseases 10th Revision (ICD-10) medical comorbidity diagnoses, antidepressant and other medications, clinical and biological markers of COVID-19 severity at baseline, and death certificates. Patient characteristics included: sex, age, and hospital, which was categorized into three classes following the administrative clustering of AP–HP hospitals in Paris and its suburbs based on their geographical location (i.e., AP–HP Centre—Paris University, Paris Saclay University, Henri Mondor University Hospitals and at home hospitalization; AP–HP Nord and Hôpitaux Universitaires Paris Seine-Saint-Denis; and AP–HP Sorbonne University); date of hospitalization, which was categorized by tertile (i.e., from 2 May 2020 to 5 December 2020; from 6 December 2020 to 15 March 2021; and from 16 March 2021 to 2 November 2021). The number of medical conditions, based on 2-digit ICD-10 diagnosis codes reported by practitioners, was recorded and categorized by sextile (i.e., 0–4, 5–7, 8–9, 10–12, 13–17, and 18+). Medications other than antidepressants included medications frequently co-prescribed with antidepressants (i.e., any benzodiazepine or Z-drug, any antipsychotic, any mood stabilizer medication), and medications used according to compassionate use or as part of a clinical trial (i.e., hydroxychloroquine, azithromycin, remdesivir, dexamethasone, molnupinavir, tocilizumab, sarilumab, bamlanivimab, and etesevimab). Dates of medication prescriptions were also recorded. To take into account medical indications of antidepressant prescription, we recorded whether patients had any ICD-10 diagnosis of psychiatric disorders (F00-F99) during the visit. Clinical severity of COVID-19 at hospital admission was defined as having at least one of the following criteria [39]: respiratory rate > 24 breaths/min or <12 breaths/min, resting peripheral capillary oxygen saturation in ambient air <90%, temperature >40 °C, or systolic blood pressure <100 mmHg. The biological severity of COVID-19 at hospital admission was defined as having at least one of the three following criteria [39,40]: high neutrophil-to-lymphocyte ratio, low lymphocyte-to-C-reactive protein ratio (both variables were dichotomized at the median of the values observed in the full sample), or plasma lactate levels higher than 2 mmol/L.

### 2.3. Antidepressant Use

Antidepressant use was defined as having an ongoing prescription of any antidepressant medication on the day of hospital admission and at least one prior prescription of the same molecule dating from the last 6 months. Antidepressant doses were extracted and converted to fluoxetine equivalents using conversion factors defined by prior work [41].

### 2.4. Study Baseline and Outcomes

The study baseline was defined as the date of hospital admission. For the first hypothesis, the outcome was the prevalence of antidepressant use in patients hospitalized with and without COVID-19. For the second hypothesis, the outcome was 28-day all-cause mortality from the study baseline in patients hospitalized with COVID-19. Patients who were discharged from the hospital before day 28 were considered to be alive.

### 2.5. Potential Mechanisms

Potential mechanisms of action of antidepressants against COVID-19 include immunomodulatory activity via sigma-1 receptor (S1R) agonism and non-S1R pathways (e.g., NF-κB, inflammasomes, TLR4, PPARγ) [42,43,44], antiviral and anti-inflammatory actions via functional inhibition of acid sphingomyelinase (FIASMA) [12,17,45,46,47], as well as serotonin modulatory [48] and anti-platelet activity [49]. To test these potential mechanisms, antidepressants were successively stratified (i) by class (SSRIs; non-SSRI antidepressants; Serotonin and Norepinephrine Reuptake Inhibitors (SNRIs); Tricyclic antidepressants (TCAs); and other antidepressants], (ii) according to the magnitude of their in vitro FIASMA activity [50] (high FIASMA activity (amitriptyline, clomipramine, fluoxetine, fluvoxamine, imipramine, maprotiline, paroxetine, sertraline, trimipramine); lower FIASMA activity (citalopram, escitalopram, venlafaxine, mianserin, mirtazapine)]) and (iii), among SSRIs, by their affinity for sigma-1 receptors (S1R) based on prior work [24,51] (high-affinity agonists (fluoxetine, fluvoxamine); intermediate-affinity agonists (escitalopram, citalopram); low-affinity agonist (paroxetine); and antagonist (sertraline)).

### 2.6. Statistical Analysis

We calculated frequencies and means (±standard deviations (SD)) of each baseline characteristic described above in patients hospitalized with or without COVID-19 and in inpatients with COVID-19 receiving or not receiving antidepressants, and compared them using standardized mean differences (SMDs). Then, we performed two main analyses. First, we used a univariate logistic regression model to compare the prevalence of antidepressant use in a matched analytic sample of patients hospitalized with or without COVID-19, using a 1:1 ratio based on age, sex, hospital, period of hospitalization, and a number of medical conditions. Second, we performed a univariate logistic regression model to examine the association of antidepressant use at baseline with 28-day mortality in a matched analytic sample of patients hospitalized with COVID-19 taking or not taking an antidepressant at baseline, using a 1:1 ratio based on age, sex, hospital, period of hospitalization, number of medical conditions, any current diagnosis of psychiatric disorders, use of other psychotropic medications, including benzodiazepines or Z-drugs, antipsychotic medications, and mood stabilizers, or any medication prescribed according to compassionate use or as part of a clinical trial, and clinical and biological markers of COVID-19 severity.

In both analyses, to reduce the effects of confounding, optimal matching [52] was used to obtain the smallest absolute distance across all clinical characteristics between patients with and without the diagnosis. In the case of unbalanced covariates (i.e., if SMD > 0.1) [53], a multivariable logistic regression model adjusting for the unbalanced covariates was also performed.

We performed sensitivity analyses to test the robustness of the two main results. First, we separately reproduced the above-mentioned analyses (i) in women and men, (ii) in younger and older patients (based on the median age of the full matched analytic samples), (iii) using two different periods of hospitalization (based on the median date of hospitalization in the full matched samples), and (iv) in patients with and without a current diagnosis of psychiatric disorder. In addition, we reproduced the two main analyses while using the Elixhauser Index instead of the number of diagnoses based on ICD-10 codes, to approximate medical comorbidity. Next, we examined those associations while considering active comparators. In the first analysis, we used statin use, a pharmacological class unrelated to SARS-CoV-2 infection risk [54]. For the second analysis, we used two active comparators that showed significant mortality reduction in randomized clinical trials among COVID-19 inpatients, including dexamethasone [55] and tocilizumab [56]. To limit the risk of immortal time bias, exposures to medications had to occur at baseline. For these analyses, patients taking both antidepressants and the active comparator were excluded from the analysis. Then, to assess the specificity of the eventual association between antidepressant use and 28-day mortality, we repeated this analysis while using urinary infection (i.e., acute pyelonephritis or cystitis) as a negative outcome. Finally, we reproduced the two main analyses while using nearest neighbor matching instead of optimal matching [57].

We performed several additional analyses. First, we reproduced the above-mentioned analyses for each individual antidepressant, selecting a priori 5 controls for each exposed case for the matched analytic samples. Second, we used logistic regression models to examine a potential dose-effect relationship between antidepressant use and 28-day mortality. Third, we explored potential underlying mechanisms by examining associations with the two main outcomes across antidepressant classes, and FIASMA and S1R classes. Finally, we examined the associations of fluoxetine, and fluoxetine or fluvoxamine, versus active comparators, including atorvastatin for the first analysis, and dexamethasone and tocilizumab for the second one.

For all associations, we performed residual analyses to assess the fit of the data, checked assumptions, and examined the potential influence of outliers [58]. E-values were used to quantify the sensitivity of the findings to unmeasured confounders in the main analyses [59]. Statistical significance was fixed a priori at a two-sided *p*-value < 0.025 (0.05/2) for the two main analyses, and at 0.05 for all sensitivity and exploratory analyses. All analyses were conducted in R software version 3.6.3 (R Project for Statistical Computing, R Core Team, Vienna, Austria). The study was performed in accordance with the Strengthening the Reporting of Observational Studies in Epidemiology (STROBE) guidelines.

## 3. Results

### 3.1. Prevalence of Antidepressant Use in Adult Patients Hospitalized with and without COVID-19

Of 388,945 adult inpatients tested for COVID-19, 3207 patients (0.8%) were excluded because of missing data (Figure 1). Of the remaining 385,738 patients, 41,293 (10.7%) had a laboratory-confirmed SARS-CoV-2 infection. In these 41,293 inpatients with COVID-19, antidepressant use was significantly less prevalent than in the matched control group of 41,293 inpatients without COVID-19 (1.9% (N = 772) versus 4.8% (N = 1988); Odds Ratio (OR) = 0.38, 95%CI = 0.35–0.41, *p* < 0.001; E-value = 4.70 (lower = 4.31)) (Figure 2, Table 1).

There were no significant between-group differences in baseline characteristics in the matched analytic sample (Appendix A). This association remained significant when using an alternative matching procedure (i.e., the nearest neighbor matching) (Appendix A), when stratifying by age, sex, and period of hospitalization, across all individual antidepressant classes and molecules, and when comparing antidepressant versus statin use as well as fluoxetine and fluoxetine or fluvoxamine use versus atorvastatin use. The magnitude of the association was significantly greater for antidepressants with high versus lower FIASMA activity, and with tricyclic versus SSRI antidepressants among antidepressants with high FIASMA activity. Conversely, this association did not significantly differ across S1R affinity classes, or across antidepressant classes among antidepressants with lower FIASMA activity (Appendix A, Table 1).

### 3.2. Antidepressant Use and 28-Day Mortality in Adult Patients Hospitalized with COVID-19

Of 41,293 inpatients with COVID-19, 31 patients (4.0% of those taking an antidepressant) were excluded because of missing information on the antidepressant dose. Of the remaining 41,262 patients, 741 patients (1.8%) received an antidepressant at baseline, with a median daily fluoxetine-equivalent dose of 30.0 mg (SD = 35.3, IQR = 19.0–49.5) (Figure 1), and all of them had a pre-illness prescription of antidepressants. Twenty-eight-day-mortality occurred in 4224 (10.2%) patients. The associations of baseline characteristics with 28-day mortality and the distribution of those characteristics according to antidepressant use are shown in Appendix A. There were no significant between-group differences according to antidepressant use in the matched analytic sample (Appendix A).

Twenty-eight-day mortality was significantly lower in patients taking an antidepressant at baseline than in those from the matched control group (12.8% (N = 95) versus 21.2% (N = 157); OR = 0.55; 95%CI = 0.41–0.72, *p* = 0.001; E-value = 3.04 (lower = 2.12)) (Table 2; Figure 3A). This association remained significant using an alternative matching procedure (i.e., the nearest neighbor matching) (Appendix A), when stratifying by age, sex, and period of hospitalization, for SSRIs and non-SSRI antidepressants, several individual antidepressants, including escitalopram, paroxetine, venlafaxine, mirtazapine, fluoxetine, and fluoxetine or fluvoxamine, and when comparing antidepressant, fluoxetine, and fluoxetine or fluvoxamine use versus use of dexamethasone and tocilizumab. Only antidepressant daily doses ≥20 mg fluoxetine-equivalents were significantly associated with reduced 28-day mortality, with a significantly greater magnitude of association for daily doses ≥40 mg versus doses <20 mg, supporting a dose-effect relationship, and that a minimal daily dose of 20 mg fluoxetine-equivalents is necessary to reach a significant protective association. This association was also significant for SSRI, non-SSRI, and high FIASMA activity classes, and significantly greater for non-SSRI than SSRI antidepressants, when prescribed at the usual antidepressant dose (i.e., 20–60 mg/day of fluoxetine equivalents) (Table 2; Figure 3). Contrariwise, this association did not significantly differ across S1R affinity classes or between high and lower FIASMA classes (Appendix A). Finally, antidepressant use was not associated with urinary infection, used as a negative outcome (Appendix A).

## 4. Discussion

In this multicenter, retrospective cohort study of 388,945 hospitalized adult patients who had been tested for COVID-19, there were two key findings. First, antidepressant use was approximately 2.5 times less prevalent in inpatients with COVID-19 than in a matched control group hospitalized without COVID-19, suggesting that the pre-illness use of these agents may be associated with reduced likelihood of hospitalization in patients infected with SARS-CoV-2. Second, among the 41,262 patients hospitalized with COVID-19, antidepressant use at baseline was significantly associated with a 45% reduced odds of 28-day mortality. Specifically, this relationship was observed for daily antidepressant doses ≥20 mg of fluoxetine-equivalents, with a significant dose-effect relationship. These associations remained significant in both men and women, younger and older patients, and in different periods of time marked by different SARS-CoV-2 variants. When examining specific classes of antidepressants, these benefits appear to be driven by antidepressants with high FIASMA activity in both analyses.

These results confirm preclinical [12,13,14,15,16,17,18], observational [19,25,26,27,28,29,34] and clinical [20,21,22,30] study findings, suggesting that certain antidepressants may be beneficial against COVID-19 at different stages of the illness. Our study extends these prior results by demonstrating that the use of antidepressants as a whole, including both SSRIs and non-SSRI antidepressants, may be less prevalent in hospitalized patients infected with SARS-CoV-2, possibly due to their protective effects against SARS-CoV-2 infection and/or against disease progression requiring hospitalization, and associated with reduced risk of death in inpatients with COVID-19.

This study also suggests mechanisms by which antidepressants may provide a protective effect against SARS-CoV-2 infection. We found that the use of antidepressants with high FIASMA activity, comprising specific SSRI and non-SSRI molecules, was significantly less prevalent in adult inpatients who tested positive versus negative for the SARS-CoV-2 in a matched analytic sample, and significantly associated with reduced mortality. These findings are in line with prior preclinical [45] and observational [28,29] study results, suggesting the utility of medications with FIASMA activity against COVID-19 disease progression, as well as studies showing that plasma levels of ceramides and enzyme activities of sphingomyelinase and ceramidase strongly correlate with disease clinical severity and inflammation markers in patients with COVID-19 [60,61,62,63,64]. Inhibition of the ASM catalyzing the formation of ceramides [50,65] by FIASMA antidepressants may result in two effects: antiviral—through the reduced formation of ceramide-enriched membrane domains that facilitate SARS-CoV-2 entry in cells, and anti-inflammatory—through the inhibition of ASM in endothelial cells and the immune system [7,12]. Based on these results, fluoxetine, which is on the World Health Organization’s Model List of Essential Medicines, has the greatest in vitro inhibitory effect on the ASM-ceramide system [50], a favorable pharmacokinetic profile [66], and is one of the best in tolerability [8,9] among SSRIs, should be considered a promising molecule to prioritize for randomized clinical trials in COVID-19 [7,11].

In contrast, when we stratified SSRIs by S1R affinity, we did not find that stronger S1R agonists provided more protection against COVID-19 compared to weaker agonists. Furthermore, our results support that non-SSRI antidepressants (which may not affect platelet activity) could be beneficial against COVID-19. Although these specific analyses may be underpowered, these findings suggest that the mechanisms involving S1R agonism and serotonin modulatory and anti-platelet activity [42,43,44] may be less central to explaining our results. However, the effect of antidepressants, especially FIASMA antidepressants, may result from complex interactions between these potential biological mechanisms. The relative importance of each of these mechanisms may also vary depending on the timing of treatment initiation and disease stage. For example, it is possible that FIASMA-related effects might be larger at an early stage of the disease, especially during the viral phase, whereas S1R agonist effects and serotonin antagonist effects may be more marked once the inflammatory phase has begun. Because patients included in this study were already taking an antidepressant at the time of the infection, it remains to be determined whether the relative contribution of these mechanisms is similar or different when the treatment is started after the infection.

Strengths of this work include the substantial sample size allowing increased statistical power compared to most prior studies, the large time frame relevant to different variants, the inclusion of a wide range of potential confounders, such as medical comorbidity and disease markers of severity, and information on the antidepressant dose. This study also has limitations. First, given the observational design, associations should not be interpreted as causal effects [67]. However, the replicability of the associations across prior studies, the significant dose-effect relationship, and the presence of biologically plausible mechanisms for explaining the observed associations reinforce the validity of our findings. Second, despite the multicenter design, our results may not be generalizable to outpatients and other countries. However, the congruence of our observations with the findings of other recent studies performed in other countries reduces this concern. Third, information about vaccination and obesity was not available. However, COVID-19 vaccination rates in people with psychiatric disorders, who are more likely to take antidepressants, may not differ from that observed in the general population [68]. Additionally, not considering obesity in our models may have biased our results towards the null hypothesis, because of its positive associations with COVID-19-related death [69] and antidepressant use [70]. Finally, the magnitude of the observed associations may be underestimated in our study given the high rate of antidepressant discontinuation in clinical outpatient settings [71]. Future studies reproducing our analyses while taking into account plasma levels of antidepressants and other medications would be beneficial [72].

## 5. Conclusions

In conclusion, antidepressant use is associated with a reduced likelihood of hospitalization in patients infected with SARS-CoV-2 and with a reduced risk of death in patients hospitalized with COVID-19. These associations were stronger for molecules with high FIASMA activity. These findings posit that prospective interventional studies of antidepressants with the highest FIASMA activity may be appropriate to help identify variant-agnostic, affordable, and scalable interventions for outpatient and inpatient therapy of COVID-19.

## Figures and Tables

**Figure 1 jcm-11-05882-f001:**
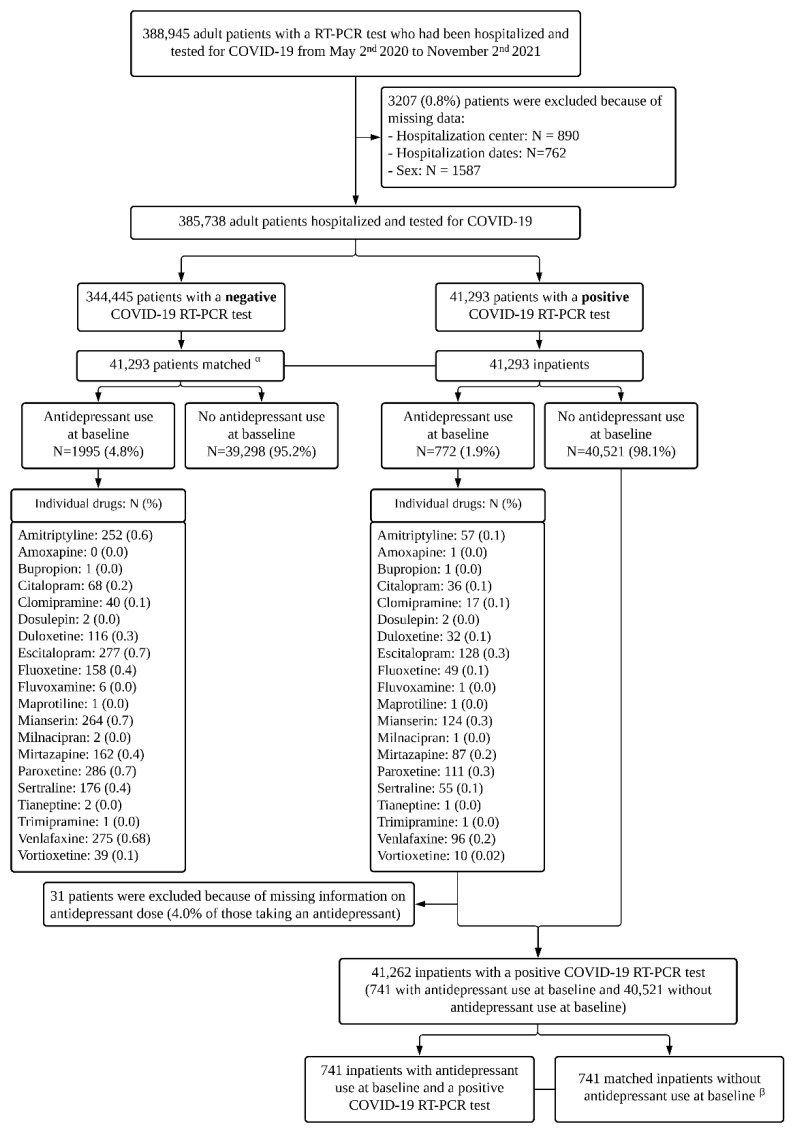
Study cohort. ^α^ Matched for age, sex, hospital, period of hospitalization, and number of medical conditions. ^β^ Matched for age, sex, hospital, period of hospitalization, number of medical conditions, any current diagnosis of psychiatric disorders, use of other psychotropic medications (benzodiazepines or Z-drugs, antipsychotic medications, mood stabilizers) or any medication prescribed according to compassionate use or as part of a clinical trial, and clinical and biological markers of COVID-19 severity.

**Figure 2 jcm-11-05882-f002:**
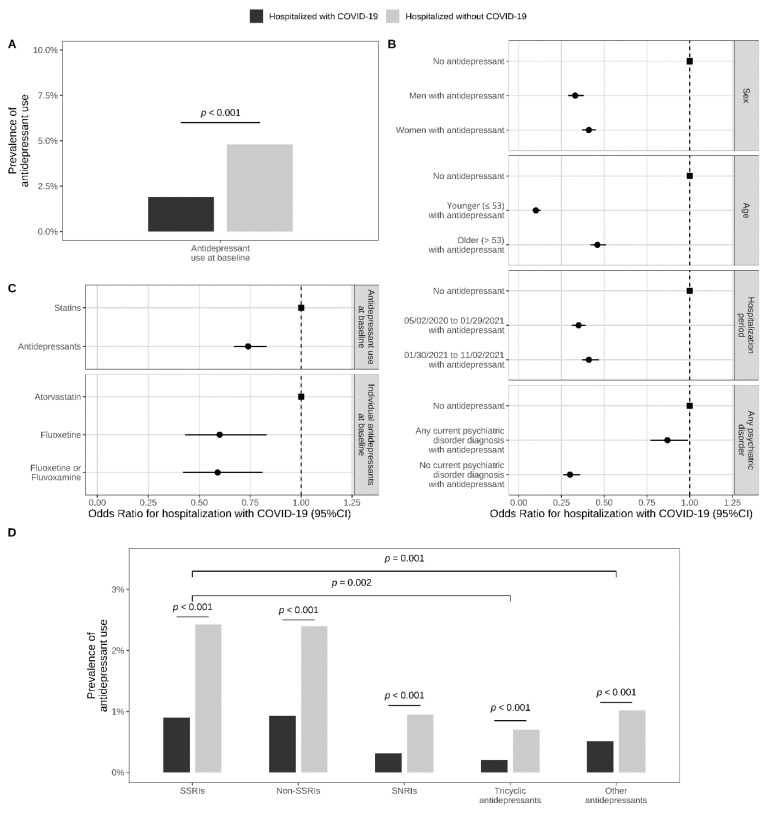
Prevalence of antidepressant use in inpatients hospitalized with and without COVID-19 (N = 82,586). (**A**) prevalence of antidepressant use in a matched analytic sample of inpatients with and without COVID-19, based on age, sex, hospital, period of hospitalization, and number of medical conditions; (**B**) association between antidepressant use and SARS-CoV-2 infection in a matched analytic sample of inpatients with and without COVID-19, stratified by age, sex, period of hospitalization, and any current diagnosis of psychiatric disorder; (**C**) comparison of antidepressant use with statin use, and fluoxetine or fluvoxamine use with atorvastatin use, in a matched analytic sample of inpatients with and without COVID-19; (**D**) associations across antidepressant classes.

**Figure 3 jcm-11-05882-f003:**
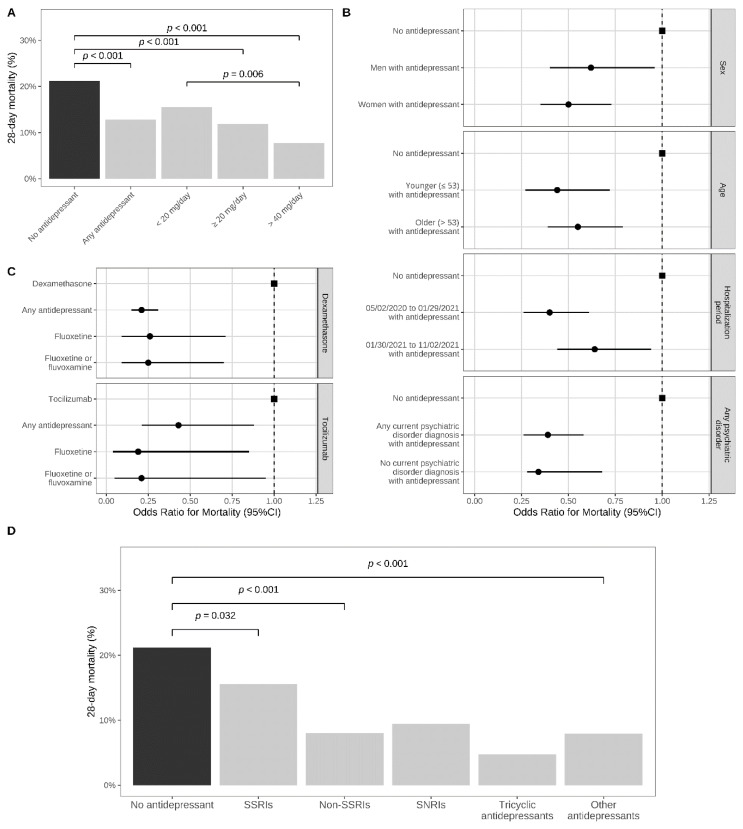
Antidepressant use and 28-day all-cause mortality in a matched analytic sample of patients hospitalized with COVID-19 (N = 1482). (**A**) Mortality rates in COVID-19 inpatients with and without an antidepressant at baseline in a matched analytic sample based on age, sex, hospital, period of hospitalization, number of medical conditions, any current diagnosis of psychiatric disorders, use of other psychotropic medications (benzodiazepines or Z-drugs, antipsychotic medications, mood stabilizers) or any medication prescribed according to compassionate use or as part of a clinical trial, and clinical and biological markers of COVID-19 severity; (**B**) associations between antidepressant use at baseline and 28-day mortality, stratified by age, sex, and period of hospitalization; (**C**) comparison of baseline use of antidepressants, fluoxetine, and fluoxetine or fluvoxamine with baseline use of dexamethasone and tocilizumab; (**D**) associations across antidepressant classes; abbreviations: ns, not significant.

**Table 1 jcm-11-05882-t001:** Prevalence of antidepressant use in a matched analytic sample of adult patients hospitalized with and without COVID-19 (N = 82,586).

	Patients Hospitalized with COVID-19 (N = 41,293)	Patients Hospitalized without COVID-19 (N = 41,293)	Hospitalized with COVID-19 versus without COVID-19 in a 1:1 Ratio Matched Analytic Sample
	N (%)	N (%)	OR (95%CI; Two-Sided *p*-Value)
No antidepressant	40,521 (98.1%)	39,298 (95.2%)	Ref.
Any antidepressant	772 (1.9%)	1988 (4.8%)	0.38 (0.35–0.41; <0.001)
Stratification by age, sex, period of hospitalization, and diagnosis of any psychiatric disorder			
Men			
Without antidepressants	20,456 (98.7%)	19,920 (96.0%)	Ref.
Any antidepressant	276 (1.33%)	821 (3.96%)	0.33 (0.29–0.38; <0.001 ***)
Women			
Without antidepressants	20,065 (97.6%)	19,378 (94.3%)	Ref.
Any antidepressant	496 (2.41%)	1174 (5.71%)	0.41 (0.37–0.45; <0.001 ***)
Younger patients (≤53)			
Without antidepressants	20,422 (99.7%)	20,289 (97.5%)	Ref.
Any antidepressant	53 (0.26%)	529 (2.54%)	0.10 (0.08–0.13; <0.001 ***)
Older patients (>53)			
Without antidepressants	20,099 (96.5%)	19,009 (92.8%)	Ref.
Any antidepressant	719 (3.45%)	1466 (7.16%)	0.46 (0.42–0.51; <0.001 ***)
Hospitalization from 2 May 2020–29 January 2021			
Without antidepressants	19,910 (98.0%)	20,081 (94.6%)	Ref.
Any antidepressant	396 (1.95%)	1149 (5.41%)	0.35 (0.31–0.39; <0.001 ***)
Hospitalization from 30 January 2021–2 November 2021			
Without antidepressants	20,611 (98.2%)	19,217 (95.8%)	Ref.
Any antidepressant	376 (1.79%)	846 (4.22%)	0.41 (0.37–0.47; <0.001 ***)
Patients with any psychiatric disorder			
Without antidepressants	3059 (88.7%)	6105 (87.3%)	Ref.
Any antidepressant	389 (11.3%)	890 (12.7%)	0.87 (0.77–0.99; 0.035 *)
Patients without psychiatric disorders			
Without antidepressants	37,462 (99.0%)	33,154 (96.7%)	Ref.
Any antidepressant	383 (1.01%)	1145 (3.34%)	0.30 (0.26–0.33; <0.001 ***)
Antidepressant classes and individual molecules			
SSRIs	388 (0.9%)	1002 (2.43%)	0.38 (0.34–0.43; <0.001 ***)
Escitalopram	128 (0.3%)	277 (0.7%)	0.46 (0.37–0.57; <0.001 ***)
Paroxetine	111 (0.3%)	286 (0.7%)	0.39 (0.31–0.48; <0.001 ***)
Sertraline	55 (0.1%)	176 (0.4%)	0.31 (0.23–0.42; <0.001 ***)
Fluoxetine	49 (0.1%)	158 (0.4%)	0.31 (0.22–0.43; <0.001 ***)
Citalopram	36 (0.1%)	68 (0.2%)	0.53 (0.35–0.79; 0.002**)
Vortioxetine	10 (0.0%)	39 (0.1%)	0.26 (0.13–0.51; <0.001 ***)
Fluvoxamine	1 (0.0%)	6 (0.0%)	0.17 (0.02–1.38; 0.097)
Fluoxetine or fluvoxamine	50 (0.1%)	164 (0.4%)	0.30 (0.22–0.42; <0.001 ***)
Non-SSRI antidepressants	384 (0.93%)	993 (2.40%)	0.38 (0.34–0.43; <0.001 ***)
SNRIs	128 (0.31%)	392 (0.95%)	0.32 (0.27–0.4; <0.001 ***)
Venlafaxine	96 (0.2%)	275 (0.7%)	0.35 (0.28–0.44; <0.001 ***)
Duloxetine	32 (0.1%)	116 (0.3%)	0.28 (0.19–0.41; <0.001 ***)
Milnacipran	1 (0.0%)	2 (0.0%)	NA
Tricyclic antidepressants	78 (0.2%)	295 (0.7%)	0.26 (0.2–0.34; <0.001 ***)
Amitriptyline	57 (0.1%)	252 (0.6%)	0.23 (0.17–0.30; <0.001 ***)
Clomipramine	17 (0.1%)	40 (0.1%)	0.42 (0.24–0.75; 0.003 **)
Dosulepin	2 (0.0%)	2 (0.0%)	NA
Maprotiline	1 (0.0%)	1 (0.0%)	NA
Trimipramine	1 (0.0%)	1 (0.0%)	NA
Amoxapine	1 (0.0%)	0 (0.0%)	NA
Imipramine	0 (0.0%)	1 (0.0%)	NA
Other antidepressants	211 (0.51%)	423 (1.02%)	0.50 (0.42–0.59; <0.001 ***)
Mianserin	124 (0.3%)	264 (0.6%)	0.47 (0.38–0.58; <0.001 ***)
Mirtazapine	87 (0.2%)	162 (0.4%)	0.54 (0.41–0.70; <0.001 ***)
Tianeptine	1 (0.0%)	2 (0.0%)	NA
Bupropion	1 (0.0%)	1 (0.0%)	NA
Number of antidepressants			
1	719 (1.74%)	1820 (4.41%)	0.38 (0.35–0.42; <0.001 ***)
2+	53 (0.13%)	175 (0.42%)	0.29 (0.22–0.40; <0.001 ***)
Comparing 2+ versus 1 antidepressant			
1	719 (1.74%)	1820 (4.41%)	Ref.
2+	53 (0.13%)	175 (0.42%)	0.77 (0.56–1.05; 0.103)
	Patients hospitalized with COVID-19	Patients hospitalized without COVID-19	Hospitalized with COVID-19 versus without COVID-19
	N (%)	N (%)	OR (95%CI; *p*-value) ^β^
Antidepressants grouped by class, FIASMA class, and S1R affinity class			
Comparing antidepressant classes ^α^	N = 709	N = 1788	
SSRIs	358 (50.5%)	902 (50.4%)	Ref.
Non-SSRI antidepressants	351 (49.5%)	886 (49.6%)	1.00 (0.84–1.19; 0.983)
SNRIs	110 (15.5%)	331 (18.5%)	0.84 (0.65–1.07; 0.161)
Tricyclic antidepressants	60 (8.5%)	245 (13.7%)	0.62 (0.45–0.84; 0.002 **)
Other antidepressants	181 (25.5%)	310 (17.3%)	1.47 (1.18–1.83; 0.001 **)
FIASMA classes ^α^	N = 41,209	N = 41,289	
No antidepressant	40,521 (98.1%)	39,298 (95.2%)	Ref.
High FIASMA activity	311 (0.8%)	1006 (2.4%)	0.30 (0.27–0.35; <0.001 ***)
Lower FIASMA activity	452 (1.1%)	985 (2.4%)	0.45 (0.40–0.51; <0.001 ***)
Comparing FIASMA classes ^α^	N = 688	N = 1693	
High FIASMA activity	266 (38.7%)	827 (48.8%)	0.66 (0.55–0.79; <0.001 ***)
Lower FIASMA activity	422 (61.3%)	866 (51.2%)	Ref.
S1R affinity classes	N = 40,910	N = 40,303	
No antidepressant	40,521 (99.0%)	39,298 (97.5%)	Ref.
High S1R affinity (agonist)	50 (0.1%)	164 (0.4%)	0.30 (0.22–0.42; <0.001 ***)
Intermediate S1R affinity	163 (0.4%)	341 (0.8%)	0.48 (0.39–0.57; <0.001 ***)
Low S1R affinity	111 (0.3%)	286 (0.7%)	0.39 (0.31–0.48; <0.001 ***)
High S1R affinity (antagonist)	65 (0.2%)	214 (0.5%)	0.30 (0.23–0.40; <0.001 ***)
Comparing S1R affinity classes ^α^	N = 387	N = 999	
High S1R affinity (agonist)	50 (12.9%)	164 (16.4%)	0.78 (0.53–1.15; 0.213)
Intermediate S1R affinity	162 (41.9%)	339 (33.9%)	1.23 (0.92–1.64; 0.164)
Low S1R affinity	111 (28.7%)	285 (28.5%)	Ref.
High S1R affinity (antagonist)	64 (16.5%)	211 (21.1%)	0.78 (0.55–1.11; 0.168)
Comparing antidepressant classes among antidepressants with high FIASMA activity ^α^	N = 256	N = 798	
SSRIs	198 (77.3%)	554 (69.4%)	Ref.
Non-SSRI antidepressants	58 (22.7%)	244 (30.6%)	0.67 (0.48–0.92; 0.015 *)
SNRIs	0 (0.0%)	0 (0.0%)	NA
Tricyclic antidepressants	58 (22.7%)	244 (30.6%)	0.67 (0.48–0.92; 0.015 *)
Other antidepressants	0 (0.0%)	0 (0.0%)	NA
Comparing antidepressant classes among antidepressants with lower FIASMA activity ^α^	N = 416	N = 549	
SSRIs	153 (36.8%)	318 (37.1%)	Ref.
Non-SSRI antidepressants	263 (63.2%)	540 (62.9%)	1.01 (0.79–1.29; 0.922)
SNRIs	84 (20.2%)	231 (26.9%)	0.76 (0.55–1.04; 0.082)
Tricyclic antidepressants	0 (0.0%)	0 (0.0%)	NA
Other antidepressants	179 (43.0%)	309 (36.0%)	1.20 (0.92–1.57; 0.172)
Antidepressant use versus statin use ^α^	N = 2063	N = 4473	
Antidepressants	772 (37.4%)	1995 (44.6%)	0.74 (0.67–0.83; <0.001 ***)
Statines	1291 (62.6%)	2478 (55.4%)	Ref.
Fluoxetine use versus atorvastatin use ^α^	N = 831	N = 1659	
Atorvastatin	782 (94.1%)	1501 (90.5%)	Ref.
Fluoxetine	49 (5.9%)	158 (9.5%)	0.60 (0.43–0.83; 0.002 **)
Fluoxetine or fluvoxamine use versus atorvastatin use ^α^	N = 832	N = 1665	
Atorvastatin	782 (94.0%)	1501 (90.2%)	Ref.
Fluoxetine or fluvoxamine	50 (6.0%)	164 (9.8%)	0.59 (0.42–0.81; 0.001 **)

The matched analytic sample of adult patients hospitalized with and without COVID-19 was built based on age, sex, hospital, period of hospitalization, and number of medical conditions. ^α^ Patients with two antidepressants or more from different classes were excluded from the analyses. * Two-sided *p* < 0.05; ** *p* < 0.01; *** *p* < 0.001. Abbreviations: SSRIs, selective serotonin reuptake inhibitors; SNRIs, Serotonin-norepinephrine reuptake inhibitors; FIASMA, functional inhibition effect on acid sphingomyelinase; S1R, Sigma-1 receptor; OR, odds ratio; CI, confidence interval; NA, not applicable.

**Table 2 jcm-11-05882-t002:** Antidepressant use and 28-day all-cause mortality in a matched analytic sample of patients hospitalized with COVID-19 (N = 1482).

	Daily Antidepressant Dose	Antidepressant Use at Baseline	Matched Control Group Not Taking an Antidepressant at Baseline (1:1 ratio)	Crude Logistic Regression in the Matched Analytic Sample	Multivariable Logistic Regression Adjusted for Unbalanced Covariates
	Median (IQR)	Deaths/ Patients (%)	Deaths/ Patients (%)	OR (95%CI; *p*-Value)	AOR (95%CI; *p*-Value)
Any antidepressant	30.0 (19.0–49.5)	95/741 (12.8%)	157/741 (21.2%)	0.55 (0.41–0.72; <0.001 ***)	-
Stratification by age, sex, period of hospitalization, and diagnosis of any psychiatric disorders					
Sex					
Women	30.4 (17.5–48.0)	54/477 (11.3%)	91/454 (20.0%)	0.51 (0.35–0.73; <0.001 ***)	0.50 (0.35–0.73; <0.001 ***) ^a^
Men	25.0 (20–50.8)	41/264 (15.5%)	66/287 (23.0%)	0.62 (0.40–0.95; 0.028 *)	0.62 (0.40–0.96; 0.031 *) ^b^
Age					
Younger patients (≤79 y)	30.0 (20.0–52.1)	25/341 (7.3%)	58/378 (15.3%)	0.44 (0.27–0.72; 0.001 **)	-
Older patients (>79 y)	25.5 (15.0–47.4)	70/400 (17.5%)	99/363 (27.3%)	0.57 (0.40–0.80; 0.001 **)	0.55 (0.39–0.79; 0.001 **) ^c^
Period of hospitalization					
2 May 2020–29 January 2021	33.0 (20.0–50.9)	38/373 (10.2%)	76/368 (20.7%)	0.44 (0.29–0.66; <0.001 ***)	0.40 (0.26–0.61; <0.001 ***) ^d^
30 January 2021–2 November 2021	25.0 (16.5–45.0)	57/368 (15.5%)	81/373 (21.7%)	0.44 (0.45–0.96; <0.001 ***)	0.64 (0.44–0.94; 0.023 *) ^e^
Psychiatric disorders					
Patients with any psychiatric disorder	35.3 (20–60)	45/388 (11.6%)	102/405 (25.2%)	0.39 (0.27–0.57; <0.001 ***)	0.39 (0.26–0.58; <0.001 ***) ^f^
Patients without any psychiatric disorder	24.1 (15.9–40.5)	50/353 (14.2%)	94/336 (28%)	0.42 (0.29–0.62; <0.001 ***)	0.44 (0.28–0.68; <0.001 ***) ^g^
Dose effect					
Fluoxetine-equivalent daily dose (mg)					
<20 mg	10.1 (6.0–11.9)	29/187 (15.5%)	157/741 (21.2%)	0.68 (0.44–1.05; 0.086)	-
≥20 mg	40.0 (23.7–60.0)	66/553 (11.9%)	157/741 (21.2%)	0.50 (0.37–0.69; <0.001 ***)	-
20 mg–60 mg	40.0 (20.0–40.0)	53/423 (12.5%)	157/741 (21.2%)	0.53 (0.38–0.75; <0.001 ***)	-
>40 mg	64.0 (50.6–81.0)	18/233 (7.7%)	157/741 (21.2%)	0.31 (0.19–0.52; <0.001 ***)	-
>60 mg	80.0 (79.1–117.5)	13/130 (10.0%)	157/741 (21.2%)	0.41 (0.23–0.75; 0.004 **)	-
Fluoxetine-equivalent daily dose (mg)					
<20 mg	10.1 (6.0–11.9)	29/187 (15.5%)	-	Ref.	-
≥20 mg	40.0 (23.7–60.0)	66/553 (11.9%)	-	0.74 (0.46–1.18; 0.208)	-
20 mg–60 mg	40.0 (20.0–40.0)	53/423 (12.5%)	-	0.78 (0.48–1.27; 0.321)	-
>40 mg	64.0 (50.6–81.0)	18/233 (7.7%)	-	0.47 (0.27–0.80; 0.006 **)	-
>60 mg	80.0 (79.1–117.5)	13/130 (10.0%)	-	0.72 (0.39–1.33; 0.289)	-
Number of antidepressants					
1	26.2 (16.0–45.0)	89/689 (12.9%)	157/741 (21.2%)	0.55 (0.42–0.73; <0.001 ***)	
2+	48.1 (27.9–74.7)	6/52 (11.5%)	157/741 (21.2%)	0.49 (0.20–1.16; 0.103)	
Comparing 2+ versus one antidepressant					
1	26.2 (16.0–45.0)	89/689 (12.9%)	-	Ref.	-
2+	48.1 (27.9–74.7)	6/52 (11.5%)	-	0.87 (0.36–2.12; 0.774)	-
	Daily antidepressant dose	Antidepressant use at baseline	Matched control group not taking an antidepressant at baseline (1:5 ratio)	Crude logistic regression in the matched analytic sample	Multivariable logistic regression adjusted for unbalanced covariates
	Median (IQR)	Deaths/ Patients (%)	Deaths/ Patients (%)	OR (95%CI; *p*-value)	AOR (95%CI; *p*-value)
Individual antidepressants					
SSRIs					
Escitalopram	30.0 (20.0–40.0)	20/123 (16.3%)	137/615 (22.3%)	0.68 (0.40–1.13; 0.139)	0.56 (0.33–0.95; 0.031 *) ^h^
Paroxetine	30.0 (20.0–40.0)	13/107 (12.1%)	132/535 (24.7%)	0.42 (0.23–0.78; 0.006)	0.43 (0.23–0.79; 0.007 **) ^i^
Sertraline	40.0 (20.0–50.0)	8/55 (14.5%)	57/275 (20.7%)	0.65 (0.29–1.46; 0.296)	0.58 (0.25–1.36; 0.210) ^j^
Fluoxetine	20.0 (20.0–40.0)	5/45 (11.1%)	61/225 (27.1%)	0.34 (0.13–0.89; 0.028 *)	0.36 (0.13–0.95; 0.040 *) ^k^
Citalopram	20.0 (20.0–40.0)	7/36 (19.4%)	39/180 (21.7%)	0.87 (0.36–2.14; 0.766)	0.72 (0.28–1.84; 0.489) ^l^
Vortioxetine	22.5 (15.0–30.0)	1/9 (11.1%)	9/45 (20%)	0.50 (0.06–4.53; 0.538)	0.45 (0.04–4.84; 0.511) ^m^
Fluvoxamine	42.0 (NA)	0/1 (0.0%)	0/5 (0.0%)	NA	NA
Fluoxetine or Fluvoxamine	20.0 (20.0–40.0)	5/46 (10.9%)	61/230 (26.5%)	0.34 (0.13–0.89; 0.029 *)	0.36 (0.13–0.96; 0.040 *) ^n^
SNRIs					
Venlafaxine	20.2 (10.1–40.5)	7/90 (7.8%)	99/450 (22%)	0.30 (0.13–0.67; 0.003 *)	0.28 (0.13–0.64; 0.002 **) ^o^
Duloxetin	40.2 (40.2–60.3)	1/30 (3.3%)	24/150 (16%)	0.18 (0.02–1.39; 0.101)	0.29 (0.03–2.48; 0.258) ^p^
Milnacipran	30.0 (NA)	0/1 (0.0%)	0/5 (0.0%)	NA	NA
Tricyclic antidepressants					
Amitriptyline	8.2 (3.4–19.0)	6/54 (11.1%)	51/270 (18.9%)	0.54 (0.22–1.32; 0.176)	0.62 (0.24–1.61; 0.328) ^q^
Clomipramine	31.5 (26.2–35.0)	3/17 (17.6%)	18/85 (21.2%)	0.80 (0.21–3.08; 0.743)	1.15 (0.27–4.87; 0.853) ^r^
Dosulepine	87.0 (NA)	0/1 (0.0%)	2/5 (40.0%)	NA	NA
Maprotiline	51.0 (NA)	0/1 (0.0%)	0/5 (0.0%)	NA	NA
Trimipramine	45.0 (NA)	0/1 (0.0%)	3/5 (60.0%)	NA	NA
Amoxapine	30.0 (NA)	0/1 (0.0%)	1/5 (20.0%)	NA	NA
Other antidepressants					
Mianserin	8.0 (4.0–12.0)	24/122 (19.7%)	139/610 (22.8%)	0.83 (0.51–1.35; 0.451)	0.66 (0.40–1.09; 0.106) ^s^
Mirtazapine	23.7 (11.9–35.6)	6/85 (7.1%)	97/425 (22.8%)	0.26 (0.11–0.61; 0.002 *)	0.21 (0.09–0.5; <0.001 ***) ^t^
Tianeptine	60.0 (NA)	0/1 (0.0%)	2/5 (40.0%)	NA	NA
Bupropion	16.5 (NA)	0/1 (0.0%)	0/5 (0.0%)	NA	NA
	Daily antidepressant dose	Antidepressant use at baseline	Matched control group not taking an antidepressant at baseline (1:1 ratio)	Crude logistic regression in the matched analytic sample	Multivariable logistic regression
	Median (IQR)	Deaths/ Patients (%)	Deaths/ Patients (%)	OR (95%CI; *p*-value)	AOR (95%CI; *p*-value) ^β^
Antidepressants prescribed at the usual fluoxetine-equivalent daily dose (20–60 mg) grouped by class, FIASMA, and S1R affinity					
Antidepressant classes ^α^		N = 387	N = 741		
SSRIs	40.0 (20.0–40.0)	39/250 (15.6%)	157/741 (21.2%)	0.69 (0.47–1.01; 0.056)	0.63 (0.41–0.96; 0.032 *)
Non-SSRI antidepressants	30.0 (23.7–40.5)	11/137 (8.03%)	157/741 (21.2%)	0.32 (0.17–0.62; 0.001 ***)	0.23 (0.12–0.47; <0.001 ***)
SNRIs	30.4 (20.2–40.5)	5/53 (9.43%)	157/741 (21.2%)	0.39 (0.15–0.99; 0.048 *)	0.39 (0.14–1.06; 0.064)
Tricyclic antidepressants	26.4 (24.8–35.0)	1/21 (4.76%)	157/741 (21.2%)	NA	NA
Other antidepressants	26.0 (23.7–47.4)	5/63 (7.94%)	157/741 (21.2%)	0.32 (0.13–0.81; 0.017 *)	0.15 (0.06–0.42; <0.001 ***)
Comparing antidepressant classes ^α^		N = 387			
SSRIs	40.0 (20.0–40.0)	39/250 (15.6%)	-	Ref.	Ref.
Non-SSRI antidepressants	30.0 (23.7–40.5)	11/137 (8.03%)	-	0.47 (0.23–0.96; 0.037 *)	0.41 (0.18–0.92; 0.031 *)
SNRIs	30.4 (20.2–40.5)	5/53 (9.43%)	-	0.56 (0.21–1.51; 0.253)	0.74 (0.24–2.26; 0.593)
Tricyclic antidepressants	26.4 (24.8–35.0)	1/21 (4.76%)	-	NA	NA
Other antidepressants	26.0 (23.7–47.4)	5/63 (7.94%)	-	0.47 (0.18–1.24; 0.125)	0.27 (0.09–0.8; 0.018 *)
FIASMA classes ^α^		N = 261	N = 741		
High FIASMA	31.5 (20.0–40.0)	20/156 (12.8%)	157/741 (21.2%)	0.55 (0.33–0.90; 0.018 *)	0.53 (0.31–0.91; 0.022 *)
Lower FIASMA	40.0 (20.0–40.0)	21/105 (20.0%)	157/741 (21.2%)	0.93 (0.56–1.55; 0.78)	0.72 (0.40–1.28; 0.262)
Comparing FIASMA classes ^α^		N = 261			
High FIASMA	31.5 (20.0–40.0)	20/156 (12.8%)	-	0.59 (0.30–1.15; 0.121)	0.71 (0.32–1.59; 0.409)
Lower FIASMA	40.0 (20.0–40.0)	21/105 (20.0%)	-	Ref.	Ref.
S1R affinity classes ^α^		N = 249	N = 741		
High S1R affinity (agonist)	20.0 (20.0–40.0)	3/30 (10.0%)	157/741 (21.2%)	0.41 (0.12–1.38; 0.151)	0.45 (0.13–1.58; 0.211)
Intermediate S1R affinity	40.0 (20.0–40.0)	19/89 (21.3%)	157/741 (21.2%)	1.01 (0.59–1.73; 0.972)	0.88 (0.47–1.63; 0.685)
Low S1R affinity	30.0 (20.0–40.0)	11/85 (12.9%)	157/741 (21.2%)	0.55 (0.29–1.07; 0.077)	0.51 (0.25–1.05; 0.068)
High S1R affinity (antagonist)	30.0 (20.0–40.0)	7/45 (15.6%)	157/741 (21.2%)	0.69 (0.3–1.56; 0.369)	0.66 (0.27–1.61; 0.358)
Comparing S1R affinity classes ^α^		N = 249			
High S1R affinity (agonist)	20.0 (20.0–40.0)	3/30 (10.0%)	-	0.75 (0.19–2.88; 0.673)	1.85 (0.71–4.86; 0.211)
Intermediate S1R affinity	40.0 (20.0–40.0)	19/89 (21.3%)	-	1.83 (0.81–4.11; 0.146)	1.01 (0.23–4.42; 0.989)
Low S1R affinity	30.0 (20.0–40.0)	11/85 (12.9%)	-	Ref.	Ref.
High S1R affinity (antagonist)	30.0 (20.0–40.0)	7/45 (15.6%)	-	1.24 (0.44–3.45; 0.682)	1.29 (0.40–4.19; 0.668)
Comparing antidepressant classes among antidepressants with high FIASMA ^α^		N = 178			
SSRIs	30.0 (20.0–40.0)	19/158 (12.0%)	-	Ref.	Ref.
Non-SSRI antidepressants	26.3 (24.8–35.0)	1/20 (5.0%)	-	NA	NA
SNRIs	NA	NA	-	NA	NA
Tricyclic antidepressants	26.3 (24.8–35.0)	1/20 (5.0%)	-	NA	NA
Other antidepressants	NA	NA	-	NA	NA
Comparing antidepressant classes among antidepressants with lower FIASMA ^α^		N = 289			
SSRIs	40.0 (20.0–40.0)	19/89 (21.3%)	-	Ref.	Ref.
Non-SSRI antidepressants	26.0 (23.7–40.5)	9/100 (9.0%)	-	0.36 (0.16–0.85; 0.020 *)	0.22 (0.07–0.69; 0.010 *)
SNRIs	30.4 (20.2–40.5)	4/39 (10.3%)	-	0.42 (0.13–1.33; 0.141)	0.6 (0.12–2.89; 0.522)
Tricyclic antidepressants	NA	NA	-	NA	NA
Other antidepressants	26.0 (23.7–47.4)	5/61 (8.2%)	-	0.33 (0.12–0.94; 0.037 *)	0.13 (0.03–0.51; 0.003 **)
	Daily antidepressant dose	Antidepressant use at baseline	Matched control group taking an active comparator at baseline (1:1 ratio)	Crude logistic regression in the matched analytic sample	Multivariable logistic regression adjusted for unbalanced covariates
	Median (IQR)	Deaths/ Patients (%)	Deaths/ Patients (%)	OR (95%CI; *p*-value)	AOR (95%CI; *p*-value)
Antidepressant use versus dexamethasone	30.0 (19.0–49.5)	53/518 (10.2%)	157/518 (30.3%)	0.26 (0.19–0.37; <0.001 *)	0.21 (0.15–0.31; <0.001 *) ^u^
Antidepressant use versus tocilizumab	23.7 (15.2–40.5)	39/306 (12.7%)	59/306 (19.3%)	0.61 (0.39–0.95; 0.028 *)	0.43 (0.21–0.88; 0.022 *) ^v^
	Daily antidepressant dose	Antidepressant use at baseline	Matched control group taking an active comparator at baseline (1:5 ratio)	Crude logistic regression in the matched analytic sample	Multivariable logistic regression adjusted for unbalanced covariates
	Median (IQR)	Deaths/ Patients (%)	Deaths/ Patients (%)	OR (95%CI; *p*-value)	AOR (95%CI; *p*-value)
Fluoxetine use versus dexamethasone	20.0 (20.0–40.0)	5/45 (11.1%)	73/225 (32.4%)	0.26 (0.1–0.69; 0.007 **)	0.26 (0.09–0.71; 0.009 **) ^w^
Fluoxetine use versus tocilizumab	20.0 (20.0–40.0)	4/44 (9.1%)	50/220 (22.7%)	0.34 (0.12–1.00; 0.049 *)	0.19 (0.04–0.85; 0.030 *) ^x^
Fluoxetine or fluvoxamine use versus dexamethasone	20.0 (20.0–40.0)	5/46 (10.9%)	74/230 (32.2%)	0.26 (0.10–0.68; 0.006 **)	0.25 (0.09–0.70; 0.008 **) ^y^
Fluoxetine or fluvoxamine use versus tocilizumab	20.0 (20.0–40.0)	4/45 (8.9%)	52/225 (23.1%)	0.32 (0.11–0.95; 0.040 *)	0.21 (0.05–0.95; 0.043 *) ^z^

The matched analytic sample of adult COVID-19 inpatients with and without antidepressant use at baseline was based on age, sex, hospital, period of hospitalization, number of medical conditions, any current diagnosis of psychiatric disorders, use of other psychotropic medications (benzodiazepines or Z-drugs, antipsychotic medications, mood stabilizers) or any medication prescribed according to compassionate use or as part of a clinical trial, and clinical and biological markers of COVID-19 severity. ^α^ Patients with two antidepressants or more from different classes were excluded from the analysis. ^β^ AOR was obtained using multivariable logistic regression models adjusted for the same variables used for building the matched analytic sample (i.e., age, sex, hospital, period of hospitalization, number of medical conditions, any current diagnosis of psychiatric disorders, use of other psychotropic medications (benzodiazepines or Z-drugs, antipsychotic medications, mood stabilizers) or any medication prescribed according to compassionate use or as part of a clinical trial, and clinical and biological markers of COVID-19 severity), and antidepressant dose (df = 19). ^a^ Adjusted for hospital and number of medical conditions. ^b^ Adjusted for hospital. ^c^ Adjusted for hospital. ^d^ Adjusted for age. ^e^ Adjusted for number of medical conditions. ^f^ Adjusted for hospital, number of medical conditions and any mood stabilizer medication. ^g^ Adjusted for age, hospital, number of medical conditions, any medication prescribed according to compassionate use or as part of a medical trial, and any antipsychotic medication. ^h^ Adjusted for age and sex. ^i^ Adjusted for hospital. ^j^ Adjusted for age, sex, hospitalization period, and biological severity of COVID-19. ^k^ Adjusted for sex and hospital. ^l^ Adjusted for age, sex, hospital, and hospitalization period. ^m^ Adjusted for sex, hospital, any medication prescribed according to compassionate use or as part of a clinical trial, and clinical severity of COVID-19. ^n^ Adjusted for sex and hospital ^o^ Adjusted for age, sex, hospital, and hospitalization period. ^p^ Adjusted for age, sex, hospital, hospitalization period, biological severity of COVID-19, and clinical severity of COVID-19. ^q^ Adjusted for age, hospitalization period, and biological severity of COVID-19. ^r^ Adjusted for age, hospital, hospitalization period, biological severity of COVID-19, and clinical severity of COVID-19. ^s^ Adjusted for age, sex, and clinical severity of COVID-19. ^t^ Adjusted for age, hospital, hospitalization period, and number of medical conditions. ^u^ Adjusted by age, sex, hospital, hospitalization period, any psychiatric disorder, any medication according to compassionate use or as part of a clinical trial (except dexamethasone), any benzodiazepine or Z-drug, biological severity of COVID-19, and clinical severity of COVID-19. ^v^ Adjusted by age, sex, hospital, hospitalization period, any psychiatric disorder, any medication according to compassionate use or as part of a clinical trial (except tocilizumab), any benzodiazepine or Z-drug, any antipsychotic medication, any mood stabilizer medication, biological severity of COVID-19, and clinical severity of COVID-19. ^w^ Adjusted by age, sex, hospital, hospitalization period, number of medical conditions, any psychiatric disorder, any medication according to compassionate use or as part of a clinical trial (except dexamethasone), any benzodiazepine or Z-drug, any mood stabilizer medication, biological severity of COVID-19, and clinical severity of COVID-19. ^x^ Adjusted by age, sex, hospital, hospitalization period, number of medical conditions, any psychiatric disorder, any medication according to compassionate use or as part of a clinical trial (except tocilizumab), any benzodiazepine or Z-drug, any mood stabilizer medication, biological severity of COVID-19, and clinical severity of COVID-19. ^y^ Adjusted by age, sex, hospital, hospitalization period, number of medical conditions, any psychiatric disorder, any medication according to compassionate use or as part of a clinical trial (except dexamethasone), any benzodiazepine or Z-drug, any mood stabilizer medication, biological severity of COVID-19, and clinical severity of COVID-19. ^z^ Adjusted by age, sex, hospital, hospitalization period, number of medical conditions, any psychiatric disorder, any medication according to compassionate use or as part of a clinical trial (except tocilizumab), any benzodiazepine or Z-drug, any mood stabilizer medication, biological severity of COVID-19, and clinical severity of COVID-19. * Two-sided *p* < 0.05; ** *p* < 0.01; *** *p* < 0.001. Abbreviations: SSRIs, selective serotonin reuptake inhibitors; SNRIs, Serotonin-norepinephrine reuptake inhibitors; FIASMA, high functional inhibition effect on acid sphingomyelinase; S1R, Sigma-1 receptors; OR, odds ratio; AOR, adjusted odds ratio; CI, confidence interval; NA, not applicable; -, irrelevant or no unbalanced covariate.

## Data Availability

Data from the AP–HP Health Data Warehouse can be obtained upon request at https://eds.aphp.fr//. The statistical code is available upon request.

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
