# Peer review of "Antidepressant Use and Its Association with 28-Day Mortality in Inpatients with SARS-CoV-2: Support for the FIASMA Model against COVID-19"

_jcm, 2022, doi:10.3390/jcm11195882_

Round 1
Reviewer 1 Report
In this manuscript, the authors reported Antidepressant use and its association with 28-day mortality in inpatients with SARS-CoV-2. The results have been presented and discussed an organized and systematic way. I found the manuscript interesting and relevant. I have no major comments, but some points to enhance the manuscript.
Minor Comments:
1) The Introduction is poorly written. The Introduction should more clearly spell out what work has been done before and what is novel in the current work.
2) Please check that all acronyms are defined on first usage both in the abstract and the main text of the manuscript.
3) Page 4, line 170 (among SSRIs, by their affinity for sigma-1 receptors (S1R) based on prior work [47]): This sentence refers a wrong reference. It is necessary to cite the reference (https://doi.org/10.1016/j.jocn.2021.03.010) instead of reference 47.
4) Page 4, line 162 (…via functional inhibition of acid sphingomyelinase (FIASMA): This sentence refers a wrong reference. It is necessary to cite the reference (https://doi.org/10.1177/00185787211073465) instead of reference 43.
Author Response
September 27th, 2022
Prof. Dr. Norihiro Kokudo
Editor-in-Chief, Journal of Clinical Medicine,
Dear Pr. Kokudo,
Thank you very much for giving us the opportunity to submit a revised version of our manuscript “Antidepressant use and its Association with 28-day Mortality in Inpatients with SARS-CoV-2: Support for the FIASMA model against COVID-19” to be considered for publication in Journal of Clinical Medicine.
We appreciate the time and effort that you and the reviewers have dedicated in providing valuable feedback on our manuscript. A point-by-point response is detailed below, and we have highlighted the corresponding changes in yellow within the manuscript.
The manuscript contains no data, patient information, or other material or results that have been published elsewhere. It has not been submitted to any other journal, nor have the results been presented at any scientific meeting.
We thank you in advance for considering this contribution.
Best regards,
Nicolas Hoertel, MD, MPH, PhD,
Corentin Celton Hospital, AP-HP.Centre, Paris University,
4 parvis Corentin Celton; 92130 Issy-les-Moulineaux, France
Phone: 0033 (0) 1 58 00 44 21
Email: nico.hoertel@yahoo.fr / nicolas.hoertel@aphp.fr
REVIEWER 1
1/ In this manuscript, the authors reported Antidepressant use and its association with 28-day mortality in inpatients with SARS-CoV-2. The results have been presented and discussed an organized and systematic way. I found the manuscript interesting and relevant. I have no major comments, but some points to enhance the manuscript.
Answer: We thank the reviewer for this positive comment.
2/ Minor Comments:
The Introduction is poorly written. The Introduction should more clearly spell out what work has been done before and what is novel in the current work.
Answer: Folloing this important comment, we rewrote parts of the introduction and insisted more on what is novel in this study, as follows:
P2: “Global spread of the different variants of Severe Acute Respiratory Syndrome Coro-navirus 2 (SARS-CoV-2) has led to an infectious disease crisis worldwide [1–4]. Because a large proportion of the world’s population is currently unvaccinated, effective treatments of Coronavirus Disease 2019 (COVID-19)—especially those that can be administered oral-ly, have good tolerability and low rate of medical contraindications [5], are inexpensive and immediately available—are urgently needed to reduce COVID-19-related mortality and morbidity [6]. This is particularly important in low- and middle-income countries, where access to vaccines and approved treatments against COVID-19 is limited [7].
Several lines of research suggests that certain well-tolerated [8,9] antidepressants, es-pecially the Selective Serotonin Reuptake Inhibitor (SSRI) medications fluvoxamine or fluoxetine, could be beneficial against COVID-19, and thus a potential mean of reaching this goal [7,10–12]. Firstly, several preclinical studies have demonstrated in vitro efficacy of several SSRI and non-SSRI antidepressants—particularly fluoxetine—against different variants of SARS-CoV-2 in human and non-human host cells [12–18]. Secondly, a retro-spective cohort study conducted in an adult psychiatric facility suggested a significant negative association of antidepressant use—particularly fluoxetine—with laborato-ry-detectable SARS-CoV-2 infection [19]. Thirdly, in the ambulatory setting, three studies [20–22], including two randomized, placebo-controlled trials (RCT) [20,22] and one non-randomized open-label clinical study [21] found a significant association between the short-term use (10-15 days) of fluvoxamine within 7 days of symptom onset and reduced risk of clinical deterioration. Contrariwise, an RCT of fluvoxamine [23] prescribed at 100 mg/d among over-weighted and obese outpatients with COVID-19 showed no significant benefit on the risk of emergency department visits, hospitalizations or death, contrasting with the findings of TOGETHER and STOP-COVID, in which fluvoxamine was prescribed at a dose of 200 and 300 mg/d, respectively. Fourthly, an observational study found that exposure to antidepressants, especially to those that functionally inhibit acid sphingomy-elinase, was associated with reduced incidence of emergency department visitation or hospital admission among SARS-CoV-2 positive outpatients, in a dose-dependent manner and from daily doses of at least 20 mg fluoxetine equivalents [24]. Fifthly, five retrospective observational cohort studies [25–29] of patients with COVID-19 in the acute-care setting reported reduced death or mechanical ventilation in those taking SSRIs, particularly fluoxetine. Of these five studies, two [25,29] reported a similar association in those taking non-SSRI antidepressants, particularly mirtazapine and venlafaxine. Finally, a prospec-tive cohort study of patients admitted in intensive care unit (ICU) for COVID-19 reported a significant association between the use of fluvoxamine for 15 days and reduced mortality [30]. Altogether, these findings suggest that the use of certain antidepressants, when pre-scribed at a dose of at least 20 mg fluoxetine equivalents, may reduce clinical deterioration of patients infected with SARS-CoV-2 in both ambulatory and acute-care settings.
However, most prior studies focused on a limited number of antidepressant mole-cules (e.g., only SSRIs). In addition, several of these studies used composite outcomes, such as intubation or death [25,28], which may prove challenging for the interpretation of results. Finally, it remains unknown whether the potential benefit of certain antidepres-sants in COVID-19 in the acute-care setting is dose-dependent and only observed beyond a certain dose threshold. This knowledge is important to help determine the best drug can-didates and their optimal dosing for future clinical trials, as well as to progress in the identification of the mechanisms underlying this potential effect.
To address these knowledge gaps, we used the Assistance Publique-Hôpitaux de Paris (AP-HP) Health Data Warehouse (‘Entrepôt de Données de Santé’ (EDS)) [25,28,29,31–38], which includes data on all adult inpatients aged 18 years or over who had been admitted to any of the 36 AP-HP Greater Paris University hospitals and tested for SARS-CoV-2 infection by a Reverse Transcription Polymerase Chain Reaction (RT-PCR) test at their admission, and performed a large (N=388,945) multicenter retro-spective cohort study.
In this study, our primary aim was two-fold: (i) to test the hypothesis that the preva-lence of antidepressant use in patients hospitalized with COVID-19 would be lower than in patients with similar characteristics hospitalized without COVID-19, and (ii) to exam-ine, among patients hospitalized with COVID-19, whether antidepressant use is associ-ated with reduced 28-day mortality. Our secondary aim was to examine whether this po-tential association could only concern specific antidepressant classes or molecules, is dose-dependent, and/or only observed beyond a certain dose threshold.”
3/ Please check that all acronyms are defined on first usage both in the abstract and the main text of the manuscript.
Answer: We carefully checked that all acronyms are defined on first usage both in the abstract and the main text of the manuscript.
4/ Page 4, line 170 (among SSRIs, by their affinity for sigma-1 receptors (S1R) based on prior work [47]): This sentence refers a wrong reference. It is necessary to cite the reference (https://doi.org/10.1016/j.jocn.2021.03.010) instead of reference 47.
Answer: We thank the reviewer for noticing this mistake, which is now corrected. We also carefully rechecked all references cited.
5/ Page 4, line 162 (…via functional inhibition of acid sphingomyelinase (FIASMA): This sentence refers a wrong reference. It is necessary to cite the reference (https://doi.org/10.1177/00185787211073465) instead of reference 43.
Answer: We thank the reviewer for noticing this mistake, which is now corrected. We also carefully rechecked all references cited.
REVIEWER 2
6/ In this multicenter, retrospective cohort study of 388,945 hospitalized adult patients who had been tested for COVID-19, study findings about “antidepressant use is associated with reduced likelihood of hospitalization in patients infected with SARS-CoV-2 and with reduced risk of death in patients hospitalized with COVID-19 .” Moreover, study findings suggests mechanisms by which antidepressants may provide a protective effect against SARS-CoV-2 infection. The study conducted by the authors and the results obtained undoubtedly contribute not only to the treatment of patients with mental disorders but also to the fight against Covid-19.The paper addresses a very interesting topic, and reviewer compliment the authors for the choice. In expressing an opinion on this manuscript, the reviewer would like to make some advise and recommendation. Therefore authors of the manuscript are kindly asked to consider the reviewer's recommendation and to provide response to each issue raised:
Answer: We warmly thank the reviewer for this positive comment on our work.
7/ Line 35: Please remove phrase (quote) “(1) Background: “
Line 37: Please remove phrase (quote) “(2) Methods: “
Line 42: Please remove phrase (quote) “(3) Results: “
Line 47-48: Please remove phrase (quote) “(4) Conclusions: “
Answer: We removed these phrase quotes accordingly.
8/ Line 117-122, (quote) “We extracted data from the electronic health record for each patient at the time of the hospitalization regarding patient demographic characteristics, hospitalization dates, laboratory test and RT-PCR test results, medication lists and medication administration data, ICD-10 medical comorbidity diagnoses, antidepressant and other medications, clinical and biological markers of COVID-19 severity at baseline, and death certificates. Patient characteristics included: sex; age; hospital (...)“ Taking into consideration and analyzing the above quoted, whether the authors of the study received written consent from all patients to use their data in the study? Please insert consent information into manuscript. If not, please insert into manuscript text on how you kept status of anonymous for all the patients' data used.
Answer: We thank the reviewer for the opportunity to clarify this important point. Followingthis comment, we added the following sentences in the method section as follows:
P3: “AP-HP clinical Data Warehouse initiatives ensure patient information and informed con-sent regarding the different approved studies through a transparency portal in accordance with European Regulation on data protection and authorization n°1980120 from National Commission for Information Technology and Civil Liberties (CNIL). This observational study using routinely collected data received full approval from the Institutional Review Board (IRB) of the AP-HP clinical data warehouse (decision CSE-20- 20_COVID19, IRB00011591). In accordance with French laws for this type of observational non-interventional research studies (“reference methodology MR-005”), patients were in-formed that their data could be used for research, but patient consent was not applicable, as this study did not contain factors necessitating it. Data were anonymized by the AP-HP clinical data warehouse team prior to the analyses.”
We warmly thank the reviewers for their help in improving our manuscript.

Reviewer 2 Report
I read with interest the manuscript entitled “Antidepressant use and its Association with 28-day Mortality 2 in Inpatients with SARS-CoV-2: Support for the FIASMA 3 model against COVID-19”. It is a highly interesting analysis with a fairly large sample size. The authors found that the use of antidepressants was associated with a decreased risk of severe manifestations and death from COVID-19. I have two comments to make:
Major comment
Antidepressant use was defined as having an ongoing antidepressant prescription of any antidepressant medication on the day of hospital admission and at least one prior prescription of the same molecule dating from the last 6 months (lines 147-149). I found no evidence that a mechanism has been implemented to verify the intake of antidepressants. If this did not occur, and because it is the exposure variable, the variable should refer to the prescription of antidepressants (no/yes) and not to their use (because the authors could not be sure that they were in fact ingested by the patients).
Minor comment
Due to the sample size, and the large number of significant results, I suggest implementing the effect size evaluation.
Reviewer 3 Report
In this multicenter, retrospective cohort study of 388,945 hospitalized adult patients who had been tested for COVID-19, study findings about “antidepressant use is associated with reduced likelihood of hospitalization in patients infected with SARS-CoV-2 and with reduced risk of death in patients hospitalized with COVID-19 .” Moreover, study findings suggests mechanisms by which antidepressants may provide a protective effect against SARS-CoV-2 infection. The study conducted by the authors and the results obtained undoubtedly contribute not only to the treatment of patients with mental disorders but also to the fight against Covid-19.The paper addresses a very interesting topic, and reviewer compliment the authors for the choice. In expressing an opinion on this manuscript, the reviewer would like to make some advise and recommendation. Therefore authors of the manuscript are kindly asked to consider the reviewer's recommendation and to provide response to each issue raised:
-
Line 35: Please remove phrase (quote) “(1) Background: “
-
Line 37: Please remove phrase (quote) “(2) Methods: “
-
Line 42: Please remove phrase (quote) “(3) Results: “
-
Line 47-48: Please remove phrase (quote) “(4) Conclusions: “
-
Line 117-122, (quote) “We extracted data from the electronic health record for each patient at the time of the hospitalization regarding patient demographic characteristics, hospitalization dates, laboratory test and RT-PCR test results, medication lists and medication administration data, ICD-10 medical comorbidity diagnoses, antidepressant and other medications, clinical and biological markers of COVID-19 severity at baseline, and death certificates. Patient characteristics included: sex; age; hospital (...)“ Taking into consideration and analyzing the above quoted, whether the authors of the study received written consent from all patients to use their data in the study? Please insert consent information into manuscript. If not, please insert into manuscript text on how you kept status of anonymous for all the patients' data used.
Author Response

(The authors gave the same response as above.)
